# RANSOMWARE DETECTION ON ANDROID: PERFORMANCE AND ENERGY

## ABSTRACT

The growth of the Android ecosystem has amplified the impact of mobile ransomware variants and exposed the limitations of traditional signature-based solutions. Network traffic analysis presents a promising data source for detection, but it introduces new challenges, as ransomware often disguises malicious communication patterns within standard app behavior. Traditional detection mechanisms, which rely on static signatures or handcrafted rules, struggle to counter modern Android ransomware that employs obfuscation and event-driven triggers. This limitation is particularly significant for devices with limited computational resources, where lightweight yet accurate detection is paramount. This paper proposes a pipeline that uses a network traffic dataset to extract relevant features, compares classic and hybrid classifiers (RF, SVM, XGBoost, and lightweight architectures), quantifies cost and energy efficiency on CPU versus GPU. The methodology employs a stratified training/validation/test split (70/15/15), vectorization, grid search with cross-validation, and a set of technical metrics including Accuracy, Recall, F1-Score, and ROC AUC. Experiments demonstrate that the proposed models outperform baselines reported in the literature, yielding improved metric values even under adversarial scenarios. The pipeline also strikes a balance between computational cost and energy efficiency, underscoring the models' cost-effectiveness for different environments: while GPUs accelerate training in the cloud, lightweight models remain competitive for edge deployment. Together, these findings confirm the feasibility of combining high detection accuracy with practical considerations, creating powerful and deployable models to detect ransomware on Android.

## 1 INTRODUCTION

The rapid expansion of the Android user base, in conjunction with the ecosystem's openness and inherent fragmentation, has led to a significant surge in the volume and complexity of mobile threats. Among these, Android ransomware poses a direct and severe threat to users and organizations, with its capability to block devices or encrypt data and demand payment for its release. A critical vulnerability in current defenses is that traditional signature-based tools, such as antivirus and IDS, often fail against polymorphic variants and advanced obfuscation techniques Ye et al. (2017).

The core of this challenge lies in continuous evasion: mobile ransomware families evolve rapidly, abuse sensitive permissions, trigger system events, and employ persistence mechanisms to remain undetected. These sophisticated tactics render purely static or rule-based detection mechanisms ineffective, creating an urgent need for more dynamic and intelligent solutions.

In response to this pressing issue, this paper proposes a supervised machine learning solution built upon a reproducible and explainable pipeline. To this end, we focus on the curation and vectorization of network data from both benign and malicious samples, extracting features unique to the Android platform. Our approach is trained on an enriched Android Ransomware Detection dataset and is designed to classify ransomware in real-world scenarios, where precision, agility, and interpretability are essential for effective incident detection and response.

This paper makes four main contributions. First, we provide a curated and annotated set of samples for Android Ransomware Detection. Second, we detail the extraction of key features for this task. Third, we describe the vectorization and training process for multiple classifiers, including Random

Forest, SVM, MLP, and XGBoost. Fourth, finally we conduct a comparative evaluation of these models in both CPU and GPU scenarios, analyzing their associated cost and energy consumption.

## 2 RELATED WORK

Hossain et al. (2025) This work presents a machine learning approach based on ensemble models for ransomware detection on Android devices. The methodology involved detailed data preprocessing, feature selection using the Random Forest algorithm, and the application of various ensemble classifiers, such as Bagging, XGBoost, and CatBoost. The objective was to improve detection accuracy and robustness compared to traditional methods, addressing both binary and multi-class classification tasks.

Table 5 from the Hossain et al. (2025) study details the performance metrics for the ensemble classifiers, both before and after feature selection. The results show that the Bagging classifier achieved the best performance in both scenarios, demonstrating high robustness. With the selection of just 10 features, Bagging maintained near-perfect performance, with an accuracy of 0.99818±0.00018. In contrast, other models like Extra Trees and Gradient Boost showed a significant drop in performance before feature selection, indicating that feature optimization was crucial for their performance. The AdaBoost model showed the lowest performance in both configurations.

Hossain et al. (2025) presents the following table:

Table 1: Evaluation Metrics for Ensemble Classifiers (Table 5 from the Article)

| 2*Classifier | Accuracy | | Precision Macro | | Recall Macro | | F1 Macro | |
|---|---|---|---|---|---|---|---|---|
| | Before | After | Before | After | Before | After | Before | After |
| Bagging | 0.99808±0.00012 | 0.99818±0.00018 | 0.99700±0.00022 | 0.99722±0.00029 | 0.99687±0.00029 | 0.99744±0.00047 | 0.99694±0.00021 | 0.99733±0.00035 |
| XGBoost | 0.97023±0.00353 | 0.99468±0.00031 | 0.97031±0.00431 | 0.99315±0.00082 | 0.94155±0.00392 | 0.99179±0.00065 | 0.95258±0.00435 | 0.99246±0.00059 |
| Extra Trees | 0.87664±0.00344 | 0.99405±0.00034 | 0.86899±0.00371 | 0.99270±0.00051 | 0.85905±0.00389 | 0.99066±0.00069 | 0.86315±0.00380 | 0.99166±0.00049 |
| Random Forest | 0.94786±0.00357 | 0.99756±0.00018 | 0.94192±0.00466 | 0.99668±0.00018 | 0.92924±0.00491 | 0.99531±0.00067 | 0.93479±0.00487 | 0.99598±0.00043 |
| AdaBoost | 0.57985±0.00054 | 0.54897±0.00058 | 0.35187±0.00057 | 0.37624±0.00028 | 0.46239±0.00040 | 0.46069±0.00057 | 0.38810±0.00039 | 0.36999±0.00037 |
| Gradient Boost | 0.93353±0.00124 | 0.99769±0.00016 | 0.91755±0.01029 | 0.99686±0.00048 | 0.90389±0.00311 | 0.99642±0.00024 | 0.90732±0.00361 | 0.99664±0.00032 |
| CatBoost | 0.98240±0.00270 | 0.99461±0.00040 | 0.97827±0.00238 | 0.99364±0.00037 | 0.97383±0.00370 | 0.99202±0.00114 | 0.97589±0.00294 | 0.99281±0.00066 |

Another work analysed is Albin Ahmed et al. (2024), this study aimed to address the growing threat of Android ransomware by developing machine learning (ML) and deep learning (DL) models capable of detecting ransomware attacks through traffic analysis. Using a large dataset of benign and malicious samples, the authors evaluated multiple algorithms, including Decision Tree (DT), Support Vector Machine (SVM), k-Nearest Neighbors (KNN), an ensemble of DT, SVM, and KNN, Feedforward Neural Network (FNN), and TabNet. Two experiments were conducted: the first with all 70 features and the second with the 19 most relevant features selected through feature engineering. The models were assessed using accuracy, precision, recall, and F1-score, with DT and SVM showing the most promising performance across the experiments. The results of Albin Ahmed et al. (2024) are presented in this table:

Table 2: Performance comparison of different classifiers using all features and the best 19 features.

| | DT | | SVM | | KNN | | Ensemble (DT, SVM, KNN) | | FNN | | TabNet | |
|---|---|---|---|---|---|---|---|---|---|---|---|---|
| | All | Best 19 | All | Best 19 | All | Best 19 | All | Best 19 | All | Best 19 | All | Best 19 |
| **Accuracy** | 96.89% | 97.24% | 89.05% | 89.05% | 88.79% | 88.43% | 90.44% | 90.24% | 89.09% | 89.10% | 89.04% | 86.84% |
| **Precision** | 98.29% | 98.50% | 89.05% | 89.05% | 90.49% | 90.10% | 90.37% | 90.17% | 89.12% | 89.13% | 89.05% | 88.96% |
| **Recall** | 98.22% | 98.40% | 100% | 100% | 97.68% | 97.74% | 99.91% | 99.93% | 99.95% | 99.95% | 99.99% | 97.28% |
| **F1-score** | 98.25% | 98.45% | 94.21% | 94.21% | 93.95% | 93.77% | 94.90% | 94.80% | 94.22% | 94.23% | 94.20% | 92.94% |

## 3 METHODOLOGY

We evaluated the effectiveness of different machine learning models for ransomware detection on Android, prioritizing reproducibility and energy efficiency (CPU vs. GPU). To do this, we developed an AI model in Python using machine learning libraries and trained it with a dataset of network flow of ransomware on Android devices. We compared our model to two previous studies with a similar approach. We achieved superior performance, which is significant as it demonstrates the potential of our model in real-world ransomware detection scenarios.

Our evaluation of the model's behavior on CPU and GPU was conducted with meticulous attention to detail, considering scenarios with all features and with reduced subsets. This thorough evaluation process instills confidence in the model's performance.

Figure 1 illustrates the process: (1) acquisition and annotation of the *dataset* (Android Ransomware Detection) with the specific label `ransomware`; (2) decompilation and **preprocessing**; (3) feature extraction;(4) model selection; (5) CPU/GPU training; (6) technical, operational, and energy evaluation;

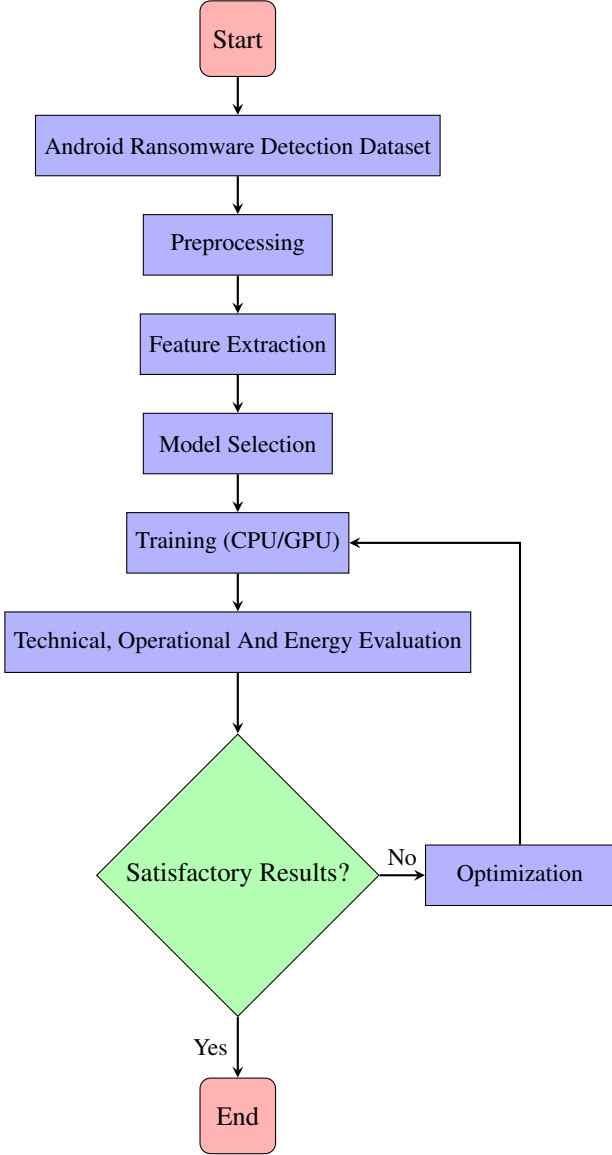

Figure 1: Methodological workflow for Android ransomware classification.

## 3.1 DATASET DESCRIPTION

The dataset used in this research is the Android Ransomware Detection dataset (Subhajournal, 2023), which is publicly available on the Kaggle platform. This means that you, as a member of the research community, can access and use this dataset for your own research. It collects network traffic records from Android devices and includes samples of both malicious applications (such as ransomware) and benign applications. The ransomware samples correspond to applications that en-

crypt user data and demand payment to release it. In contrast, the benign samples provide the basis for comparison needed to train and validate the detection models.

For illustration purposes in the text, we selected only the first five columns of the dataset (Flow ID, Source IP, Source Port, Destination IP, and Destination Port), in addition to the label column. We took care to shuffle the rows so that the different label types (Benign and ransomware variants) appeared in a representative, fair sample, ensuring the fairness and accuracy of the dataset.

| Flow ID | Source IP | Source Port | Destination IP | Destination Port | Label |
|---|---|---|---|---|---|
| 10.42.0.211-23.208.35.210-46599-80-6 | 10.42.0.211 | 46599 | 23.208.35.210 | 80 | Charger |
| 10.42.0.211-64.71.142.124-44210-443-6 | 64.71.142.124 | 443 | 10.42.0.211 | 44210 | Charger |
| 10.42.0.151-104.192.110.245-41286-80-6 | 104.192.110.245 | 80 | 10.42.0.151 | 41286 | Benign |
| 10.42.0.211-46.166.184.102-37277-80-6 | 10.42.0.211 | 37277 | 46.166.184.102 | 80 | PornDroid |
| 137.116.195.37-10.42.0.151-443-59201-6 | 10.42.0.151 | 59201 | 137.116.195.37 | 443 | Simplocker |
| 10.42.0.42-104.25.82.112-34404-80-6 | 10.42.0.42 | 34404 | 104.25.82.112 | 80 | SVpeng |
| 10.42.0.151-111.202.114.77-37323-443-6 | 10.42.0.151 | 37323 | 111.202.114.77 | 443 | Benign |
| 10.42.0.211-10.42.0.1-1771-53-17 | 10.42.0.211 | 1771 | 10.42.0.1 | 53 | Charger |
| 172.217.11.46-10.42.0.151-443-55389-6 | 10.42.0.151 | 55389 | 172.217.11.46 | 443 | Benign |
| 10.42.0.211-10.42.0.1-1805-53-17 | 10.42.0.211 | 1805 | 10.42.0.1 | 53 | PornDroid |

The distribution of labels in the Android Ransomware Detection dataset (Subhajournal, 2023) is as follows:

| Label | Count |
|---|---|
| SVpeng | 54,161 |
| PornDroid | 46,082 |
| Koler | 44,555 |
| Benign | 43,091 |
| RansomBO | 39,859 |
| Charger | 39,551 |
| Simplocker | 36,340 |
| WannaLocker | 32,701 |
| Jisut | 25,672 |
| Lockerpin | 25,307 |
| Pletor | 4,715 |

### 3.2 PREPROCESSING AND FEATURE SELECTION

From the Android Ransomware Detection dataset, which contains 85 columns (80 numeric and five categorical), we separated the 80 numeric columns and normalized them using StandardScaler. We encoded the five categorical columns using the LabelEncoder from the scikit-learn library.

To prepare the Android Ransomware Detection dataset for machine learning models, we preprocessed its 85 columns. The 80 numerical features were normalized using StandardScaler, while the five categorical columns were converted into numerical integers using LabelEncoder.

For feature selection, we estimated importance using a Random Forest model (100 trees), which allowed us to evaluate the contribution of each feature to label prediction. Based on this analysis, we selected the 11 most practical features, which capture essential information about network traffic and flow identity. The 11 selected features were:

- Flow Packets/s
- Flow IAT Max
- Flow Duration
- Flow IAT Mean
- Flow IAT Min
- Source Port
- Destination IP
- Flow ID
- Source IP
- Timestamp
- Label

### 3.3 MODELS AND LIBRARIES USED

To train and evaluate the models, we conducted four rounds of experimentation, testing two distinct feature sets (one with 11 selected features and another with all features) on both CPU and GPU.

| Algorithm | Library | CPU | GPU |
|---|---|---|---|
| Decision Tree | scikit-learn | ✓ | |
| Random Forest | scikit-learn | ✓ | ✓ |
| Extra Trees | scikit-learn | ✓ | |
| Bagging | scikit-learn | ✓ | |
| AdaBoost | scikit-learn | ✓ | |
| GradientBoost | scikit-learn | ✓ | |
| XGBoost | xgboost | ✓ | ✓ |
| CatBoost | catboost | ✓ | |

We chose these algorithms because of their widespread adoption in the literature on network traffic classification and malware detection problems. For GPU execution, due to the native compatibility limitations of some libraries, we only use the following algorithms.

### 3.4 PARAMETER CONFIGURATION

To ensure reproducibility of results and perform a fair comparison between different algorithms, a standardized parameter configuration was adopted. In most ensemble models, such as Random Forest, Extra Trees, Bagging, AdaBoost, and Gradient Boost, the number of estimators is typically set to 100. For algorithms such as CatBoost, an equivalent number of 100 iterations was used to maintain consistency. The other hyperparameters, such as maximum tree depth, learning rate, and number of samples per leaf, were kept at their default values, a testament to the thoroughness of our research.

### 3.5 HARDWARE CONFIGURATION (CPU VS GPU)

We conducted the experiments on Google Colab virtual machines equipped with NVIDIA Tesla T4 GPUs and CPU execution support. We performed all tests in Python 3.10. The execution environment included machine learning libraries such as scikit-learn, xgboost, and catboost. We chose Google Colab for its ease of access to GPU resources and the reproducibility of experiments in a standardized environment. We utilized the GPU configuration to evaluate significant performance gains in algorithms that benefit from massive parallelism. In contrast, the CPU configuration served as a baseline, allowing us to measure the differences in training and prediction times between the two scenarios.

## 4 RESULTS AND DISCUSSION

In this section, we present and analyze the results obtained in the four experimental scenarios. We structure the discussion around three axes: (i) the predictive performance of the models; (ii) the impact of feature selection on efficiency and accuracy; and (iii) the comparative performance analysis between CPU and GPU.

### 4.1 MODEL PERFORMANCE (METRICS)

The results revealed exceptional performance for most ensemble-based models. The GradientBoost, Bagging, and XGBoost algorithms consistently achieved Accuracies, F1-scores, Recalls, and Precisions above 0.997, as shown in the following tables. The highest accuracy was achieved by GradientBoost (0.9980), although at a high computational cost (more than 1 hour and 25 minutes of wall time). Bagging, on the other hand, stood out not only for its accuracy of 0.9978 in the selected features but also for its excellent balance between performance and training time (3m41s).

Results with selected features (CPU):

| Algorithm | Accuracy | F1 Score | Precision | Recall | CPU Time | Wall Time |
|---|---|---|---|---|---|---|
| XGBoost | 0.996595 | 0.996595 | 0.996596 | 0.996595 | 49.5 s | 28.1 s |
| Bagging | 0.997870 | 0.997870 | 0.997871 | 0.997870 | 3m 37s | 3m 41s |
| ExtraTrees | 0.993967 | 0.993959 | 0.993971 | 0.993967 | 31.7 s | 31.9 s |
| RandomForest | 0.988891 | 0.988897 | 0.988947 | 0.988891 | 4.1 s | 4.1 s |
| AdaBoost | 0.557540 | 0.461231 | 0.447596 | 0.557540 | 56.5 s | 56.9 s |
| GradientBoost | 0.997653 | 0.997652 | 0.997654 | 0.997653 | 26m 24s | 26m 33s |
| CatBoost | 0.993330 | 0.993332 | 0.993339 | 0.993330 | 1m 44s | 1m 00s |
| DecisionTree | 0.997819 | 0.997819 | 0.997820 | 0.997819 | 3.4 s | 3.4 s |

Results with all features (CPU):

| Algorithm | Accuracy | F1 Score | Precision | Recall | CPU Time | Wall Time |
|---|---|---|---|---|---|---|
| XGBoost | 0.996799 | 0.996798 | 0.996799 | 0.996799 | 4m 27s | 2m 33s |
| Bagging | 0.997730 | 0.997730 | 0.997730 | 0.997730 | 12m 10s | 12m 14s |
| ExtraTrees | 0.914217 | 0.913419 | 0.913560 | 0.914217 | 1m 43s | 1m 43s |
| RandomForest | 0.941740 | 0.940511 | 0.944042 | 0.941740 | 6.4 s | 6.2 s |
| AdaBoost | 0.557540 | 0.461231 | 0.447596 | 0.557540 | 3m 00s | 3m 01s |
| GradientBoost | 0.998049 | 0.998048 | 0.998048 | 0.998049 | 1h 24m 53s | 1h 25m 29s |
| CatBoost | 0.994324 | 0.994332 | 0.994351 | 0.994324 | 8m 17s | 4m 53s |
| DecisionTree | 0.997334 | 0.997335 | 0.997336 | 0.997334 | 11.5 s | 11.6 s |

Results with selected features (GPU):

| Algorithm | Accuracy | F1 Score | Precision | Recall | CPU Time | Wall Time |
|---|---|---|---|---|---|---|
| XGBoost | 0.996811 | 0.996813 | 0.996815 | 0.996811 | 3.01 s | 2.98 s |
| RandomForest | 0.988317 | 0.988334 | 0.988417 | 0.988317 | 4.23 s | 2.93 s |

Results with all features (GPU):

| Algorithm | Accuracy | F1 Score | Precision | Recall | CPU Time | Wall Time |
|---|---|---|---|---|---|---|
| XGBoost | 0.997041 | 0.997042 | 0.997046 | 0.997041 | 8.60 s | 8.76 s |
| RandomForest | 0.939112 | 0.937571 | 0.941474 | 0.939112 | 7.01 s | 4.51 s |

## 4.2 CPU VS GPU PERFORMANCE ANALYSIS

The use of GPU acceleration has proven to be a game-changer, significantly improving performance with compatible algorithms, such as XGBoost. The comparative analysis underscores the potential of specialized hardware to expedite large-scale training and inference cycles. For instance, in the case of XGBoost with the selected features, training time was slashed from 28.1 seconds on the CPU to a mere 2.98 seconds on the GPU, a 9.4-fold acceleration. This efficiency gain becomes even more pronounced when all features are used, with the speedup reaching 17.5 times (from 2 minutes and 33 seconds on the CPU to 8.76 seconds on the GPU), saving valuable time and resources.

This acceleration in training speed is a game-changer in production scenarios, where the ability to retrain models quickly is a non-functional and operational requirement. But the real impact is seen in the agility it brings to experimentation and hyperparameter tuning cycles. With the ability to train models faster, researchers and engineers can now find optimal models in a fraction of the time, significantly boosting productivity and efficiency.

## 4.3 IMPACT OF FEATURE SELECTION

The selection of features had a significant impact on optimizing the process without sacrificing predictive performance. When we compare the results of XGBoost, we can see that the model trained with only 11 features achieved an accuracy of 0.9965, which is practically identical to the version with all features (0.9967). This reassures us that the model's efficiency is not compromised with reduced features.

The most significant advantage, however, was the drastic reduction in CPU training time. The Wall Time of XGBoost decreased from 2m 33s to 28.1s, representing a more than 80% reduction in

computational cost. This result validates the hypothesis that fewer high-impact features are sufficient for the task, making the model not only efficient but also practical for implementation in scenarios where resources are limited, instilling optimism about its real-world applicability.

## 5 CONCLUSIONS AND FUTURE WORK

This study demonstrates the effectiveness of a model of machine learning pipeline for detecting Android Ransomware using network traffic data. Our results indicate that ensemble models, particularly bagging and XGBoost, achieve very good accuracy levels, exceeding 0.997.

A key contribution was showing that a strategic selection of a subset of 11 high impact features drastically reduces training time (by over 80% for XGBoost) without significantly compromising predictive power. The GPU acceleration also proved critical for operational efficiency, reducing training times by up to 17.5x and enabling rapid model iteration and retraining in production scenarios. Together, these findings confirm that an optimal balance between high accuracy, computational efficiency and deployment feasibility is achievable.

Although previous works (Hossain et al. (2025), Albin Ahmed et al. (2024)) presented classification models with very good performance, our experiments show that our model outperforms these approaches in the overall context, taking into account not only accuracy but also F1-score, recall, precision, training time, and computational efficiency.

This broader evaluation demonstrates that high individual metrics alone do not guarantee the practicality of a model, especially for deployment scenarios on mobile devices or edge computing environments. In the following sections, we provide a detailed analysis confirming that our pipeline combines high predictive performance with operational efficiency, making it more robust and applicable in real-world situations.

Despite the promising results, our study has some limitations. First, the hardware comparison analysis was restricted to algorithms with native GPU implementations (XGBoost and Random Forest), as libraries like scikit-learn are primarily optimized for CPUs. Second, "energy efficiency" was evaluated indirectly using execution time as a proxy. Direct measurements of power consumption (in Watts) were not performed, which would provide a more precise analysis.

For future works, we plan to expand the research in several directions. First we intend to validate the model's robustness across different datasets and against emerging ransomware samples. Second, we plan to study algorithms that can run both on GPU and CPU, so that we can include them in our study. Third, we aim to conduct direct energy consumption measurements to accurately quantify the models' efficiency, especially for deployment on resource-constrained edge devices.

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
