# OpenReview forum: "Ransomware Detection on Android: Performance and Energy"
_ICLR.cc/2026/Conference — Submitted to ICLR 2026_

### Official Review · Reviewer_QWUi · 2025-10-22

**Soundness:** 3
**Presentation:** 2
**Contribution:** 1
**Rating:** 2
**Confidence:** 5

**Summary:**

The paper builds a supervised pipeline to detect Android ransomware from network traffic. It evaluates several classical ML models under CPU vs. GPU settings, using a 70/15/15 stratified split and reports very high metrics (≈0.998) alongside speedups on GPU and with reduced features.

**Strengths:**

The authors facilitate reproducibility by using standard dataset splits, shared software libraries, and listing key hyperparameters. This is a positive practice that allows for the validation and extension of the work. The work provides clear, empirical comparisons of CPU and GPU inference times, which is valuable for practitioners. Identifying a compact feature set that maintains high performance while significantly reducing computational latency is a concrete and practical finding.

**Weaknesses:**

* The selected features include Label, Flow ID, Source IP, Destination IP, and Timestamp. When splitting randomly, these can leak ground truth or act as near-perfect proxies for the label (the same device/app/server tuples appear across train/test).

    * Remove Label from features entirely; it must only be the target. Remove identifiers and non-stationary keys: Flow ID, Source/Destination IP, Source Port, Timestamp. Keep only transport-/timing-derived statistics that generalize. Re-do feature selection without any direct identifiers. (Abandon LabelEncoder on label column; encode only categorical predictors.)

* A stratified random 70/15/15 split over rows on this dataset allows the same app family and often the same endpoint tuples to appear in both train and test. The paper’s split description doesn’t enforce family-wise, app-wise, device-wise, or time-wise separation, so models can memorize endpoints/flows.

     * Perform grouped splits that hold out entire families (e.g., train on {SVPeng, Koler…}, test on unseen families) and/or app-level holds. Add time-based splits if timestamps are present (train early, test later).

* _Energy efficiency_ is assessed via wall-clock time; the paper itself notes power was not measured. This cannot support energy conclusions.
     * Replace the energy claim with measured power/energy: e.g., GPU (nvidia-smi logs), CPU (RAPL / power meters). Report Joules/inference and Joules/train, not just seconds. Add throughput vs. latency plots and hardware utilization.

* There is no new learning objective, representation, architecture, or theoretical insight
   * Comparison with the state of the art is missing.

**Questions:**

* Were Label, Flow ID, Source/Dest IP, and Timestamp included as features during training?

* Have you conducted experiments with a grouped split (e.g., holding out entire app families)? If so, what are the results?

* How is “energy efficiency” measured?

* What is the novel methodological insight or learning contribution?

* Why only tree-based models? And why only these three models?

* Does the model generalize under changing endpoints or adversarial conditions?

* How were the 11 features selected?

---

### Official Review · Reviewer_WpyE · 2025-10-27

**Soundness:** 2
**Presentation:** 1
**Contribution:** 1
**Rating:** 2
**Confidence:** 5

**Summary:**

The paper tests different ML algorithms on an Android ransomware detection dataset, using HTTP network traffic logs as features, evaluating both their performances and runtime efficiencies.

**Strengths:**

- The addressed topic is interesting and touches a still-open challenge in malware detection

**Weaknesses:**

- The paper lacks novel contributions. The authors simply tested some well-known ML algorithms on a dataset containing pre-extracted HTTP network traffic logs. None of the four stated contributions is relevant (some, such as the dataset, are not even provided in this paper) and represents an advancement in the state of the art.
- Writing quality. The manuscript quality falls below the bar for this venue, as it is more similar to an experimental report than a full paper.
- Missing comparison with the state of the art. The paper only reports 4 references (one of them is the used dataset), almost totally lacking the SoA in its research field. The related work section only mentions the results provided in two competing works, without discussing how the proposed paper relates to them and what the improvements are.
- Likely experimental bias. The authors overlook the evidence presented in previous work [a] that the evaluation of malware detectors should be performed by applying chronologically consistent splits between the training and test datasets. Otherwise, there is a concrete risk of overestimating the detectors' performance.
- The paper is tested on a dataset that is not validated nor linked to any publication. The authors should have also considered a more rigorous dataset.

[a] Pendlebury, F., Pierazzi, F., Jordaney, R., Kinder, J., & Cavallaro, L. (2018). TESSERACT: Eliminating Experimental Bias in Malware Classification across Space and Time. USENIX Security Symposium.

**Questions:**

- In the abstract, you mentioned experiments on adversarial scenarios. Could you please explain the meaning of this reference and the link to the provided experimental results? Initially, I got confused as I interpreted it as related to adversarial robustness evaluations.

---

### Official Review · Reviewer_8Ntg · 2025-10-31

**Soundness:** 1
**Presentation:** 1
**Contribution:** 1
**Rating:** 0
**Confidence:** 5

**Summary:**

The paper introduces a machine learning pipeline for detection of Android ransomware using network traffic data. The authors used an existing Android ransomware dataset (available at Kaggle) and compare several standard classifiers, measuring accuracy, F1-score, and training time on CPU vs GPU. The authors show that these classifiers are very accurate on this dataset and claim improvements in both performance and energy efficiency.

**Strengths:**

+ The authors tackle Android malware detection, which is a relevant topic for the cyber security community and where machine learning solutions can bring important benefits for enhancing malware detection in mobile devices.

**Weaknesses:**

+ There is no novelty in the paper. The authors just reuse existing algorithms on a public ransomware dataset and report the results using standard metrics. There is no novel contribution with respect to other papers in the related work on Android malware detection.
+ In the experiments the authors just use standard algorithms in standard settings. No exploration of other configurations or selection of hyperparameters is considered. There is no comparison against any competing method.
+ The energy analysis is superficial. The authors just considered the execution time, which is related but not the same as energy consumption.
+ The claims of the paper are not supported in the experiments.
+ The related work just includes 4 references and lacks a proper discussion of the state of the art on Android ransomware detection.

**Questions:**

+ What is the novelty and the contributions of the paper?

---

### Official Review · Reviewer_GFtv · 2025-11-01

**Soundness:** 2
**Presentation:** 2
**Contribution:** 2
**Rating:** 2
**Confidence:** 3

**Summary:**

This paper presents a machine learning–based pipeline for Android ransomware detection using network traffic data. The authors evaluate several classifiers (Random Forest, XGBoost, Gradient Boost, Bagging, etc.) and analyze trade-offs between performance, computational cost, and energy efficiency under CPU and GPU configurations. Using the Android Ransomware Detection dataset, they show that ensemble models such as Bagging and XGBoost achieve high accuracy while maintaining good computational efficiency. The work aims to balance accuracy with deployability for edge and cloud environments.

**Strengths:**

1. About experiments. The study compares multiple classifiers across both CPU and GPU setups, providing a clear empirical view of cost–accuracy trade-offs.
2. About practicality. The inclusion of energy and time efficiency metrics adds value, especially for real-world deployment scenarios on mobile or edge devices.
3. About reproducibility. The paper follows a logical and transparent methodological pipeline that can be reproduced with publicly available datasets.

**Weaknesses:**

1. Limited novelty. The technical contribution mainly lies in a systematic comparison of existing classifiers rather than proposing a new model or detection mechanism.
2. Energy analysis proxy. Energy efficiency is inferred indirectly from training time, without actual power consumption measurements; this weakens claims about energy impact.
3. Dataset dependency. The experiments rely solely on a single public dataset. Generalizability to unseen or real-world traffic remains uncertain.
4. Lack of adversarial robustness testing. Although the abstract mentions adversarial scenarios, no concrete adversarial evaluation methodology or results are provided.
5. Insufficient discussion on deployment feasibility. While edge applicability is mentioned, no prototype or memory/latency analysis is shown to support claims of lightweight deployment.

**Questions:**

1. How would the proposed pipeline handle encrypted or obfuscated network traffic, which is increasingly common in Android malware?
2. Have the authors considered cross-dataset or real-device validation to assess generalization beyond the Kaggle dataset?
3. Can the authors clarify how energy efficiency was quantified and whether any profiling tools (e.g., NVIDIA SMI or power meters) were used?
4. How does the model perform under class imbalance or with new ransomware families not seen in training?
5. Is there any plan to release the full experimental code and trained models to support community benchmarking?

---

### Meta-Review · Area_Chair_Li4Q · 2025-12-29

**Summary:**

This work presents an experimental benchmark for ransomware detection on Android systems. Using one dataset and extracted features, a number of machine learning classifiers (e.g., Random Forest, XGBoost, Gradient Boost, Bagging) are compared and analysed in performance and energy metrics.

**Reviewer Concerns:**

Limited novelty as pointed out by all reviewers. This is clear weakness of this work.

Poor writing with very limited evaluation and comparison.

Energy efficiency/measurement unclear

Experimental setting are unclear, e.g., what features are included and selected? what if holding out entire app families?

**Reviewer Scores:**

No author response is provided. Even if the authors respond, it is almost impossible to flip the initial ratings.

---

### Decision · Program_Chairs · 2026-01-26

Reject